# Three corrections for overshoot effect improved the dose for step-and-shoot intensity-modulated radiation therapy

**Chun-Yen Yu**[1,2], **Shih-Wen Wan**[3], **Yih-Chyang Weng**[4], **Ching-Han Hsu**[1]*

**1** Department of Biomedical Engineering and Environmental Sciences, National Tsing Hua University, Hsinchu, Taiwan, Republic of China, **2** Department of Medical Imaging and Radiological Sciences, Chung Shan Medical University, Taichung, Taiwan, Republic of China, **3** Department of Health Physics Division, Institute of Nuclear Energy Research, Taichung, Taoyuan, Taiwan, Republic of China, **4** Department of Radiation Oncology,Nantou Hospital, Ministry of Health and Welfare, Nantou, Taiwan, Republic of China

* cghsu@mx.nthu.edu.tw

**Data Availability Statement:** All relevant data are within the manuscript and its Supporting Information files.

## Abstract

We measured the overshoot effect in a linac and reduced it using block correction, reverse-sequence correction, and index correction. A StarTrack detector was used on a Varian iX. Five segments, $1 \times 10$ cm$^2$ in area, were designed; the centers were at −4, −2, 0, 2, and 4 cm on the $x$ axis for measuring the overshoot effect on a $10 \times 10$ cm$^2$ collimator setting. Block correction was applied to two segments. The first was on the new first segment at −6 cm, and the other was on the new last segment at 6 cm. Both two new segments were obtained from the $10 \times 10$ cm$^2$ collimator setting. The order of segments was reversed in reverse-sequence correction. Reverse-sequence correction averages the dose at every segment after two irradiations. When we used MLC Shaper, index correction reduced the first segment's index (cumulative radiation occupation) by 60% and increased the last segment's radiation occupation by 60% in a new MLC.log file. As for relative dose, the first segment had an overdose of 52.4% and the last segment had an underdose of 48.6%, when irradiated at 1 MU at 600 MU/min. The relative doses at the first segment, irradiated at 1 MU, after block correction, reverse-sequence correction, and index correction were applied decreased from 152.5% to 95.1%, 104.8%, and 100.1%, respectively. The relative doses at the last segment, irradiated at 1 MU, after block correction, reverse-sequence correction, and index correction were applied increased from 48.6% to 97.3%, 91.1%, and 95.9%, respectively. The overshoot effect depended on the speed of irradiation. High irradiation speeds resulted in notable overdosing and underdosing at the first and last segments, respectively. The three corrections mitigated the overshoot effect on dose. To save time and effort, the MLC.log file should be edited with a program in the future.

## Introduction

The multileaf collimator (MLC) is a crucial component in modern techniques, such as intensity-modulated radiation therapy (IMRT) and volumetric-modulated arc therapy (VMAT).

**Funding:** The authors received no specific funding for this work.

**Competing interests:** The authors have declared that no competing interests exist.

After MLC position is detected by the dose servo in the linac, radiation can be triggered 65 ms later [1]. Accurate dose delivery is difficult because of an overshoot effect caused by the delay in the dose servo control. Dose delivery accuracy is also affected by smaller MU and higher dose rates [2–5]. For stop-and-shoot IMRT, the overshoot effect causes an overdose at the first segment and an underdose at last segment.

Relative to other outputs, Ezzell et al. determined larger differences between planned and measured doses at per-segment outputs <1 MU. Almost 75% of the segments were skipped for quality assurance filming. Kuperman et al. formulated a method that swapped the first segment with the last [6]. They also managed to swap the first two segments with the last two. Grigorov [7] determined the dose to be >60% and <60% at the first and last segments, respectively, with 600 MU/min and 1 MU delivered for each segment, consistent with Ezzell and Kuperman (2003). Therefore, in the MLC log file, they changed the dose by +60% and −60% for the first and last segments, respectively, thereby compensating for the overshoot effect. Zhen et al [8] added two closed segments, one before the first segment and another after the last segment. Thus, when the overshoot effect is constrained to take effect at the two closed segments, the doses at the other segments are constrained to be in accordance with the treatment plan.

In this study, we measured overshoot effect on a linac. We reduced overshoot effect using three corrections and analyzed three corrections. The three corrections were called the block correction, reverse-sequence correction, and index correction.

## Materials and methods

A StarTrack (IBA Dosimetry, Schwarzenbruck, Germany) was applied on a Varian iX with 6 MV and 120 multileaf collimators (MLCs). In the StarTrack, 453 air-vented ionization chambers were arranged in five parallel $y$-axis lines, one $x$-axis line, and two diagonal lines ($27 \times 27$ cm$^2$). The distance of adjacent ionization chambers was 5 mm for the five parallel $y$-axis lines and one $x$-axis line, and 7 mm for the two diagonal lines. The Source-Axis-Distance (SAD) was always kept at 100 cm from the StarTrack's surface with a 1.5-cm solid phantom, and the collimator was set as $10 \times 10$ cm$^2$ for all measurements.

To measure the overshoot effect, we designed five segments using the software MLC SHAPER (Varian Medical Systems, Palo Alto, CA). Each segment's size was $1 \times 10$ cm$^2$, with their centers at −4, −2, 0, 2, and 4 cm on the $x$ axis (Figs 1–5). The occupation of radiation for each segment was 20%. The total dose of each irradiation was 5–50 MU, at 5-MU increments. Thus, every segment received 1 to 10 MUs at 1-MU increments. The dose rate was 100–600 MU/min, at 100-MU/min increments.

After measuring the overshoot effect, we used the three aforementioned corrections to reduce the overshoot effect. Two segments were added to the five segments as block correction. One segment was added as the new first segment before the first of five segments at −6 cm on the $x$ axis. The other segment was added as the new last segment after the last of five segments at 6 cm. Therefore, the block correction involved "1 + 5 + 1" segments. The two added segments irradiated out of the $10 \times 10$ cm$^2$ collimator. The radiation from the two added segments were blocked with the collimator.

During irradiation, the five segments were acted on sequentially from the first to fifth segments. The dose of the first segment was over, and the dose of the fifth segment was under. During the subsequent irradiation, the five segments were acted on sequentially from the fifth to first segments. The dose of the fifth segment was over, and the dose of the first segment was under. Therefore, the doses at the first and fifth segments were compensated for after two irradiations. Reverse-sequence correction averages the dose at every segment after two irradiations.

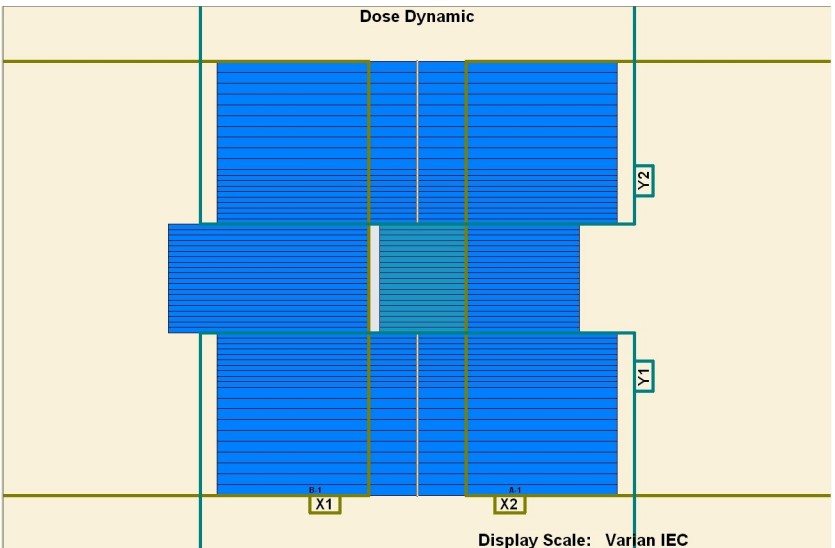

**Fig 1. Five segments designed to be 1 × 10 cm².** (1) First segment's center: −4 cm. (2) Second segment's center: −2 cm. (3) Third segment's center: 0 cm. (4) Fourth segment's center: 2 cm. (5) Fifth segment's center: 4 cm.

Because of the overshoot effect, the dose at the first segment increased by approximately 60%, and the dose at the last segment decreased at approximately 60%. In new MLC.log file, index correction reduced the first segment's index (cumulative radiation occupation) by 60% and increased the last segment's radiation occupation by 60%. Although the new MLC.log file was executed out and overshoot effect also occurred, the overshoot effect compensated for the reduced index for the first segment and increased index for last segment.

## Results

The results for the overshoot effect are presented in Figs 6–15. The $x$ axis represents the segment position, and the $y$ axis represents the relative dose. All doses were normalized to 10% of

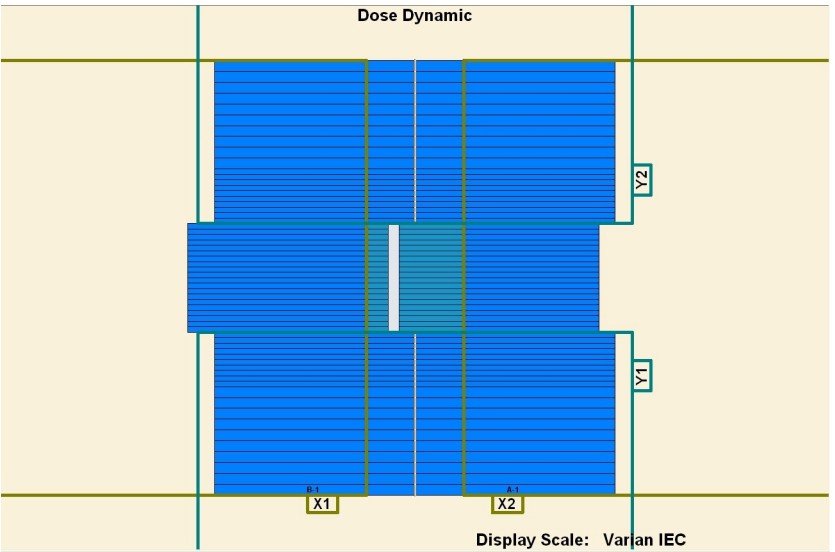

**Fig 2. Five segments designed to be 1 × 10 cm².** (1) First segment's center: −4 cm. (2) Second segment's center: −2 cm. (3) Third segment's center: 0 cm. (4) Fourth segment's center: 2 cm. (5) Fifth segment's center: 4 cm.

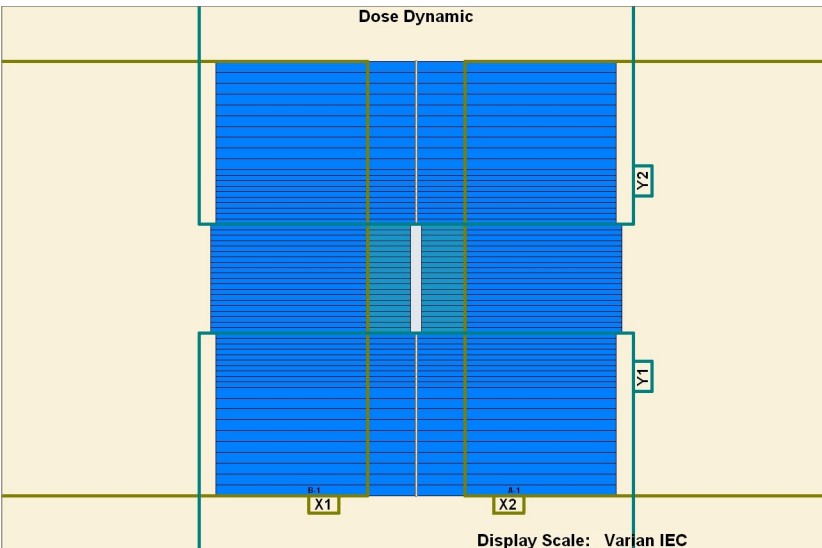

**Fig 3. Five segments designed to be 1 × 10 cm².** (1) First segment's center: −4 cm. (2) Second segment's center: −2 cm. (3) Third segment's center: 0 cm. (4) Fourth segment's center: 2 cm. (5) Fifth segment's center: 4 cm.

the third segment's, which was irradiated at 10 MU at a 100-MU/min dose rate, as presented in Fig 6. The data in Fig 7 were normalized to 20% of the third segment's, which was irradiated at 10 MU at a 100-MU/min dose rate. Data for the other irradiations were normalized to be its own fraction of the third segment's, which was irradiated at 10 MU at a 100-MU/min dose rate. The black, purple, blue, green, orange, and red lines in Figs 6–15 represent data for irradiations at the dose rates of 100, 200, 300, 400, 500, and 600 MU/min, respectively.

Table 1 lists all relative doses at each first segment. The overshoot effect was absent for the segments irradiated at 3–10 MU at the dose rates of 100 MU/min and 200 MU/min, overdose due to the overshoot effect was much more obvious for the other irradiative conditions,

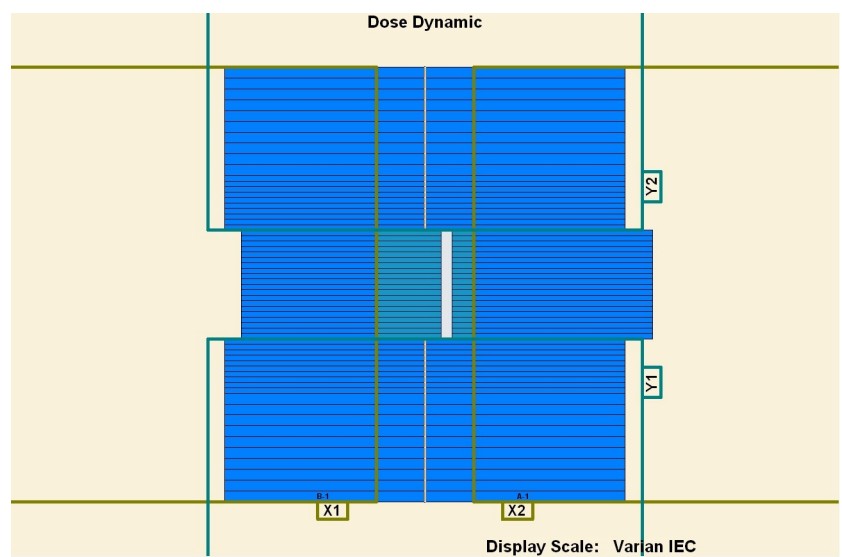

**Fig 4. Five segments designed to be 1 × 10 cm².** (1) First segment's center: −4 cm. (2) Second segment's center: −2 cm. (3) Third segment's center: 0 cm. (4) Fourth segment's center: 2 cm. (5) Fifth segment's center: 4 cm.

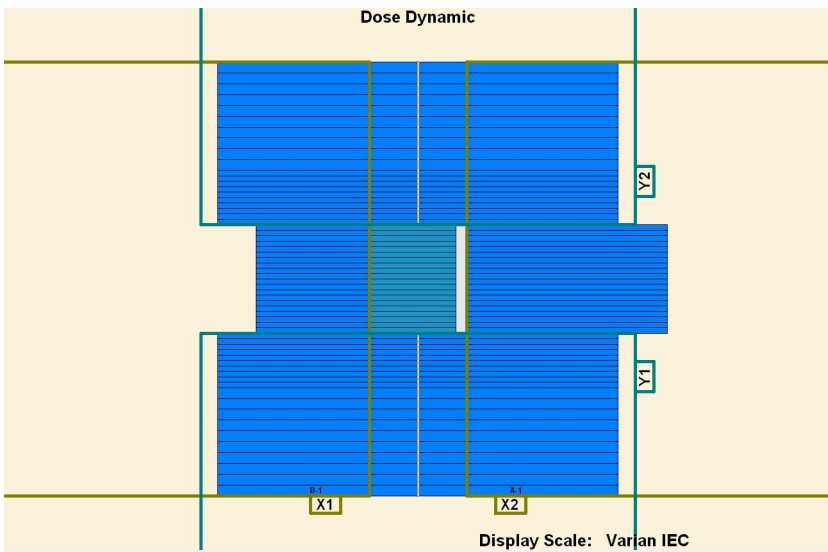

**Fig 5. Five segments designed to be 1 × 10 cm².** (1) First segment's center: −4 cm. (2) Second segment's center: −2 cm. (3) Third segment's center: 0 cm. (4) Fourth segment's center: 2 cm. (5) Fifth segment's center: 4 cm.

reaching 52.4% for irradiation at 1 MU at a 600-MU/min dose rate. The results given in Table 1 are consistent with the results of Grigorov [7], who determined the dose to be >60% and <60% at the first and last segments, respectively, with 600 MU/min and 1 MU delivered for each segment. Irradiation at 1 MU or at 600 MU/min resulted in a higher likelihood of overdose for the first segment (Table 1).

Table 2 lists the relative doses at each last segment. The underdoses in the last segment due to the overshoot effect were also obvious. In particular, 48.6% of the lowest relative doses were irradiated at 1 MU at 600 MU/min. The relative doses irradiated at 10 MU or at a 100-MU/min dose rate were much closer to 100% (at 97%) compared with other degrees of irradiation. In contrast to irradiation at 10 MU or at a 100-MU/min dose rate, the segment irradiated at 1 MU or at a 600-MU/min dose rate had a lower relative dose.

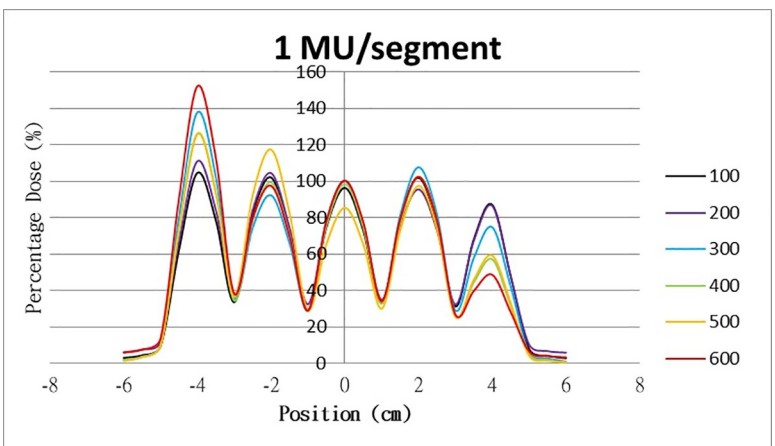

**Fig 6. Overshoot effect measured at various MU values and dose rates.** Overshoot effect, at various dose rates, for (6) 1 MU, (7) 2 MU, (8) 3 MU, (9) 4 MU, (10) 5 MU, (11) 6 MU, (12) 7 MU, (13) 8 MU, (14) 9 MU, (15) and 10 MU.

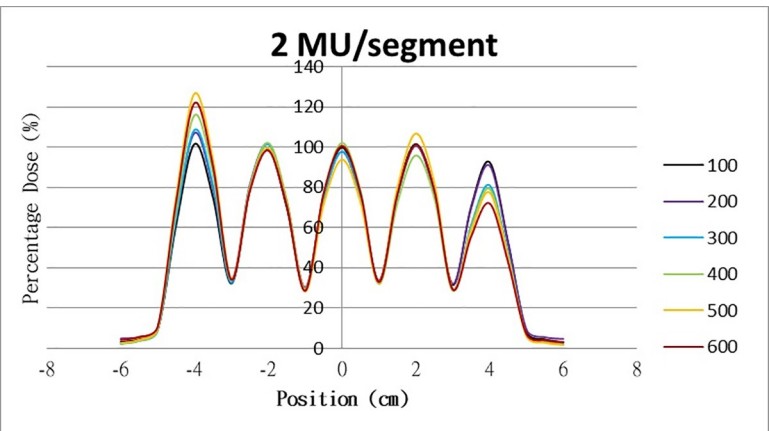

**Fig 7. Overshoot effect measured at various MU values and dose rates.** Overshoot effect, at various dose rates, for (6) 1 MU, (7) 2 MU, (8) 3 MU, (9) 4 MU, (10) 5 MU, (11) 6 MU, (12) 7 MU, (13) 8 MU, (14) 9 MU, (15) and 10 MU.

Results for the overshoot effect are presented in Figs 6 to 15, Tables 1 and 2. The overshoot effect affected the dose at the first and last segments greatly when these segments were irradiated at a smaller MU value or a higher dose rate. As such, each segment was irradiated at 1 MU, 2MU, and 3 MU at a 600-MU/min dose rate with three corrections: the block correction, reverse-sequence correction, and index correction. The results for the three corrections, before and after their application, are presented in Figs 16–18 shown. In Figs 16–18, the red, blue, orange, and green lines indicate the relative dose of the overshoot effect with no correction, block correction, reverse-sequence correction, and index correction, respectively. Table 3 lists the relative doses at the first and last segments. Specifically, for the first segment, the relative doses at 1 MU when block correction, reverse-sequence correction, and index correction were applied, decreased from 152.5% to 95.1%, 104.8%, and 100.1%, respectively. For the last segment, the relative doses at 1 MU after the aforementioned corrections were applied increased from 48.6% to 97.3%, 91.1% and 95.9%, respectively.

## Discussion

The overshoot effect was not discussed with regard to dynamic sliding window IMRT; it was only detected for step-and-shoot IMRT. We believe that the overshoot effect still affects the

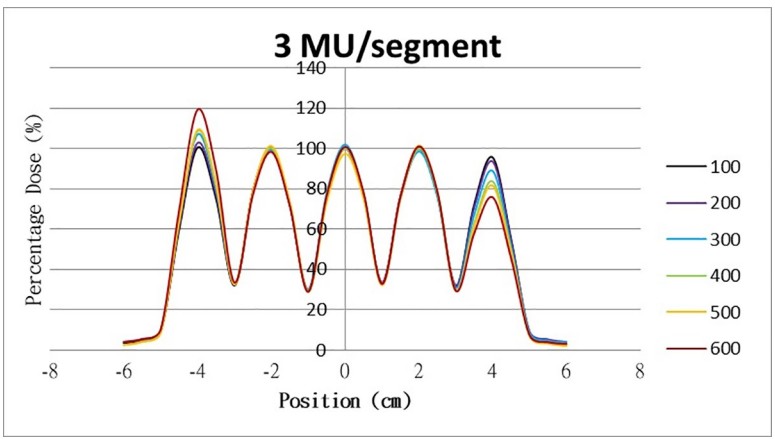

**Fig 8. Overshoot effect measured at various MU values and dose rates.** Overshoot effect, at various dose rates, for (6) 1 MU, (7) 2 MU, (8) 3 MU, (9) 4 MU, (10) 5 MU, (11) 6 MU, (12) 7 MU, (13) 8 MU, (14) 9 MU, (15) and 10 MU.

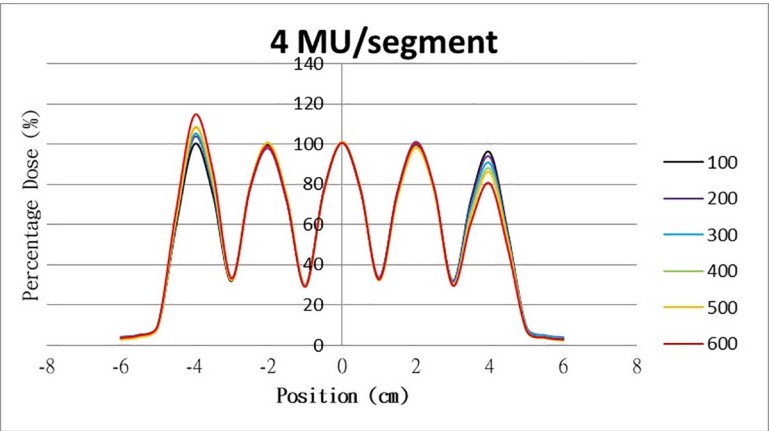

**Fig 9. Overshoot effect measured at various MU values and dose rates.** Overshoot effect, at various dose rates, for (6) 1 MU, (7) 2 MU, (8) 3 MU, (9) 4 MU, (10) 5 MU, (11) 6 MU, (12) 7 MU, (13) 8 MU, (14) 9 MU, (15) and 10 MU.

dose in dynamic sliding window IMRT during the first and last 65 ms. This is because the dose servo in the linac is still operational during the dynamic sliding window IMRT. Detection is difficult in dynamic sliding window IMRT due to constant irradiation. The presence of the overshoot effect in dynamic sliding window IMRT may also be because the segments in a single field often number more than 100, making the fraction of the dose at the first and last segments much smaller, which, in turn, greatly reduces the overshoot effect on dose. Thus, overshoot is difficult to detect in dynamic sliding window IMRT.

The overshoot effect also depends on the speed of irradiation (Figs 6 to 15). Faster irradiation speeds results in overdose and underdose at the first and last segments, respectively. Specifically, because irradiation speed depends on irradiation MU and dose rate, small MU values require high irradiation speeds to maintain the delivery of radiation, resulting in greater overdose and underdose at the first and last segments, respectively. Similarly, higher dose rates require faster irradiation speeds to maintain the delivery of radiation, resulting in greater overdose and underdose at the first and last segments, respectively. Fortunately, the overshoot effect can be reduced at slower dose rates, such as 100 MU/min and 200 MU/min. The other words, increasing minimum segment MU and limiting total number of segments per beam

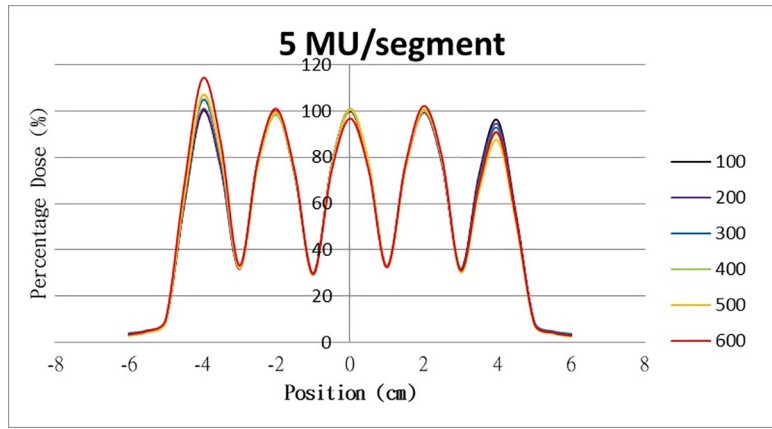

**Fig 10. Overshoot effect measured at various MU values and dose rates.** Overshoot effect, at various dose rates, for (6) 1 MU, (7) 2 MU, (8) 3 MU, (9) 4 MU, (10) 5 MU, (11) 6 MU, (12) 7 MU, (13) 8 MU, (14) 9 MU, (15) and 10 MU.

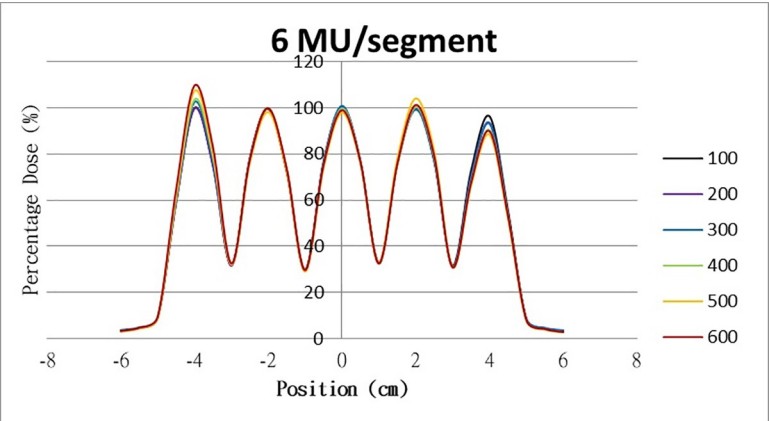

**Fig 11. Overshoot effect measured at various MU values and dose rates.** Overshoot effect, at various dose rates, for (6) 1 MU, (7) 2 MU, (8) 3 MU, (9) 4 MU, (10) 5 MU, (11) 6 MU, (12) 7 MU, (13) 8 MU, (14) 9 MU, (15) and 10 MU.

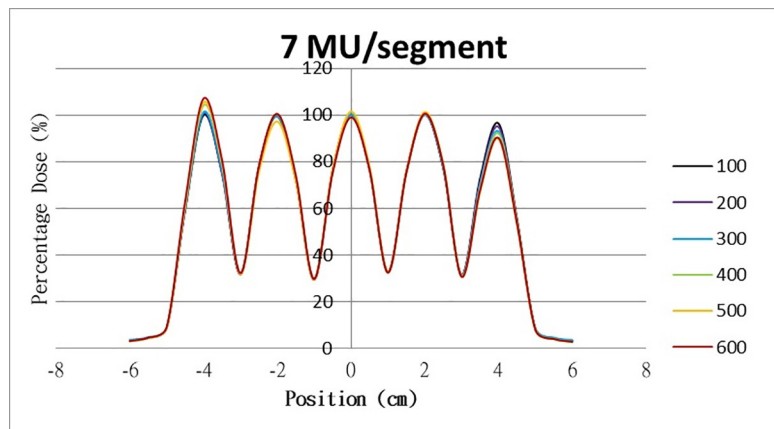

**Fig 12. Overshoot effect measured at various MU values and dose rates.** Overshoot effect, at various dose rates, for (6) 1 MU, (7) 2 MU, (8) 3 MU, (9) 4 MU, (10) 5 MU, (11) 6 MU, (12) 7 MU, (13) 8 MU, (14) 9 MU, (15) and 10 MU.

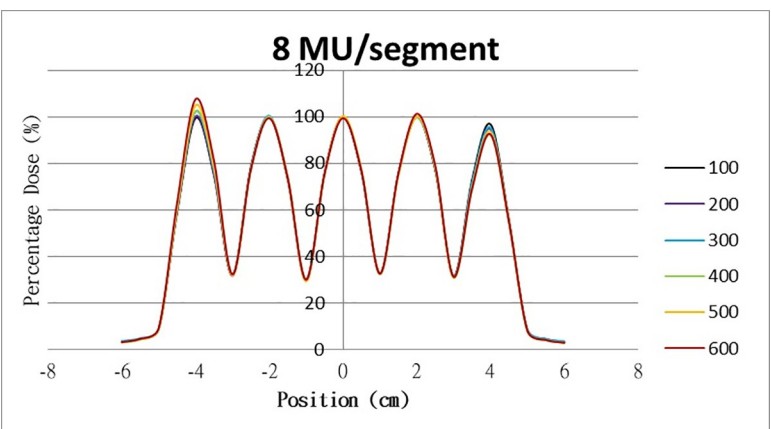

**Fig 13. Overshoot effect measured at various MU values and dose rates.** Overshoot effect, at various dose rates, for (6) 1 MU, (7) 2 MU, (8) 3 MU, (9) 4 MU, (10) 5 MU, (11) 6 MU, (12) 7 MU, (13) 8 MU, (14) 9 MU, (15) and 10 MU.

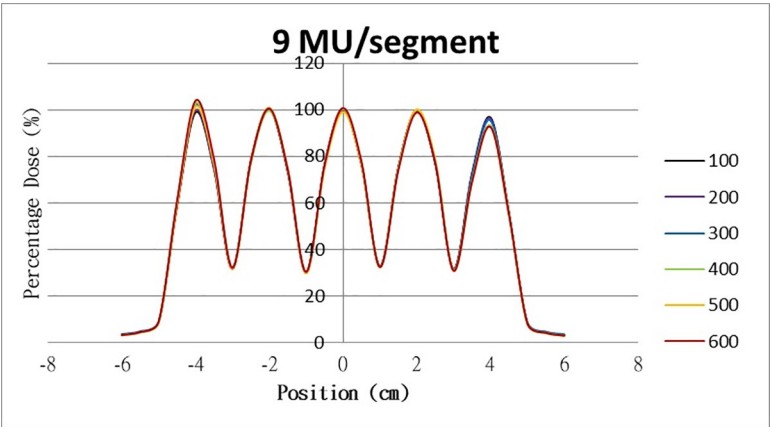

**Fig 14. Overshoot effect measured at various MU values and dose rates.** Overshoot effect, at various dose rates, for (6) 1 MU, (7) 2 MU, (8) 3 MU, (9) 4 MU, (10) 5 MU, (11) 6 MU, (12) 7 MU, (13) 8 MU, (14) 9 MU, (15) and 10 MU.

(increase minimum segment MU) during plan optimization are also reduce overshoot effect. For clinical plans which have multiple fields, very few segments, and low MUs, overshoot effect might be significant and lower GAMMA passing rate might be found.

In addition to overdose and underdose, a segment was ignored randomly during our tests when a segment was irradiated at <1 MU. The ignored dose may be delivered in the next segment. This phenomenon was also mentioned in Ezzell's article [3].

The three corrections greatly mitigated overdose and underdose. The relative dose at the first segment, when irradiated at 1 MU at 600 MU/min, was reduced from 152.5% to 98.7%, 93.4%, and 100.1% when block correction, reverse-sequence correction, and index correction were applied, respectively. The relative dose at the last segment increased from 48.6% to 97.3%, 91.1%, and 95.9% when the three aforementioned corrections were applied, respectively. For block correction, the new first segment and the new last segment were set out of field. However, setting the two new segments as closed will yield better results, as per the method proposed by Zhen et al (2016). Block correction is limited by the fact that the transmission of MLC in the field would be increased when the new two segments are irradiated. Thus, the radiation occupations of the two new segments were set to be much smaller, closer

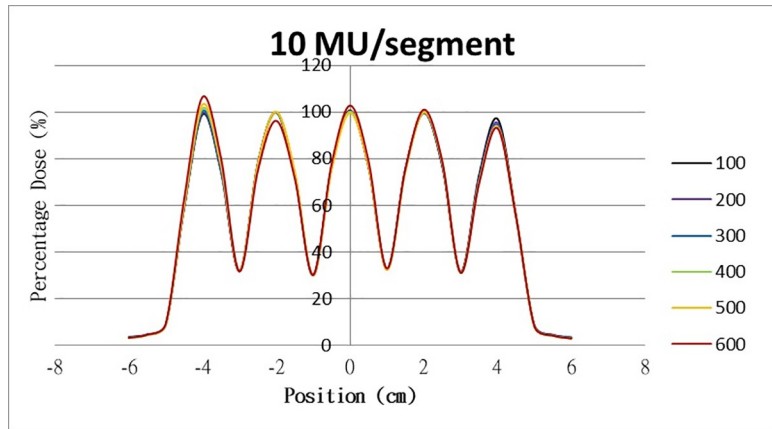

**Fig 15. Overshoot effect measured at various MU values and dose rates.** Overshoot effect, at various dose rates, for (6) 1 MU, (7) 2 MU, (8) 3 MU, (9) 4 MU, (10) 5 MU, (11) 6 MU, (12) 7 MU, (13) 8 MU, (14) 9 MU, (15) and 10 MU.

**Table 1. Relative doses at the first segment in relation to the overshoot effect.**

|  | 100 | 200 | 300 | 400 | 500 | 600 |
|---|---|---|---|---|---|---|
| 1 MU | 104.6% | 110.9% | 137.8% | 125.8% | 126.7% | 152.4% |
| 2 MU | 101.5% | 107.0% | 108.6% | 115.9% | 126.6% | 122.0% |
| 3 MU | 100.4% | 102.7% | 107.0% | 108.6% | 109.2% | 119.3% |
| 4 MU | 100.0% | 103.7% | 105.0% | 107.9% | 108.4% | 114.5% |
| 5 MU | 100.0% | 100.7% | 105.0% | 106.7% | 106.9% | 114.2% |
| 6 MU | 100.0% | 99.7% | 102.5% | 103.6% | 107.3% | 109.7% |
| 7 MU | 100.0% | 100.7% | 101.3% | 104.2% | 105.3% | 107.1% |
| 8 MU | 99.4% | 100.2% | 102.2% | 102.4% | 104.9% | 107.6% |
| 9 MU | 98.9% | 99.8% | 102.2% | 102.7% | 101.2% | 104.0% |
| 10 MU | 99.0% | 99.5% | 100.4% | 101.5% | 103.2% | 106.5% |

**Table 2. Relative doses at the last segment in relation to the overshoot effect.**

|  | 100 | 200 | 300 | 400 | 500 | 600 |
|---|---|---|---|---|---|---|
| 1 MU | 87.0% | 86.4% | 72.8% | 74.6% | 57.1% | 48.6% |
| 2 MU | 92.5% | 90.8% | 82.7% | 81.1% | 79.4% | 77.5% |
| 3 MU | 95.5% | 93.4% | 89.6% | 88.9% | 83.6% | 81.6% |
| 4 MU | 96.0% | 93.7% | 91.4% | 90.6% | 87.9% | 86.1% |
| 5 MU | 96.0% | 94.3% | 93.6% | 93.3% | 90.1% | 87.4% |
| 6 MU | 96.3% | 93.5% | 93.9% | 93.1% | 90.3% | 88.3% |
| 7 MU | 96.4% | 94.9% | 94.0% | 92.9% | 92.1% | 89.7% |
| 8 MU | 96.8% | 94.6% | 95.8% | 95.3% | 92.0% | 93.6% |
| 9 MU | 96.7% | 96.1% | 95.9% | 95.4% | 93.3% | 92.5% |
| 10 MU | 97.0% | 95.3% | 95.4% | 94.4% | 93.2% | 93.7% |

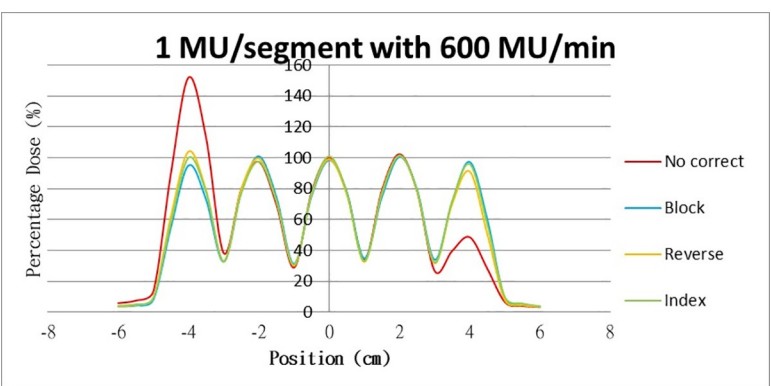

**Fig 16. Relative doses for overshoot after the three corrections were applied.** Relative doses at (16) 1 MU, 600 MU/min; (17) 2 MU, 600 MU/min; and (18) 3 MU, 600 MU/min.

to zero, to reduce the transmission as much as possible. Block correction will not work if the radiation occupations of the new two segments are set to zero; the MLC computer will not function, and the segment of radiation occupation will be zero. From our data, overdose and underdose fractions varied depending on MU and dose rate. Thus, these fractions are difficult

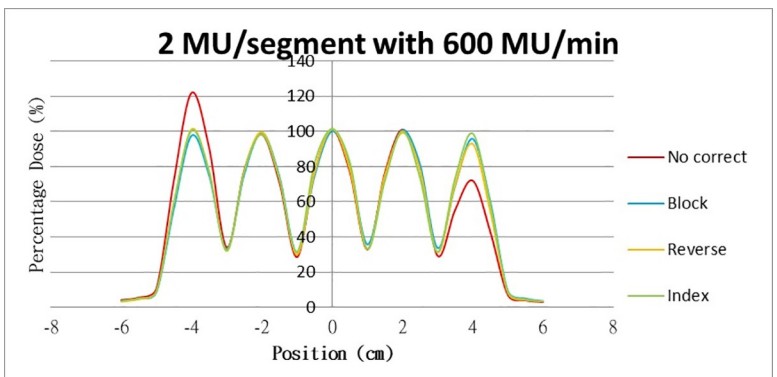

**Fig 17. Relative doses for overshoot after the three corrections were applied.** Relative doses at (16) 1 MU, 600 MU/min; (17) 2 MU, 600 MU/min; and (18) 3 MU, 600 MU/min.

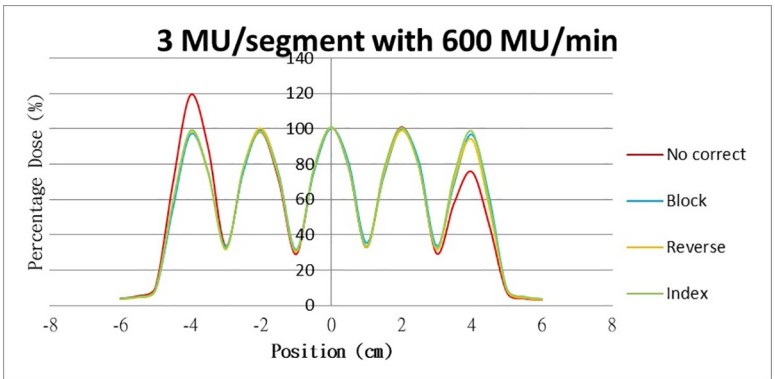

**Fig 18. Relative doses for overshoot after the three corrections were applied.** Relative doses at (16) 1 MU, 600 MU/min; (17) 2 MU, 600 MU/min; and (18) 3 MU, 600 MU/min.

to predict for a treatment field using index correction. Compared with block correction and index correction, reverse-sequence correction results in greater time savings when the MLC. log file was edited. Index recalculation and renormalization in the MLC.log file were much easier for reverse-sequence correction relative to its counterparts.

The MLC.log file had to be edited according to two principles before the three corrections were applied. The first was that the index of last segment must be 1 in the MLC.log file. The other was that every segment must be irradiated at the same MU as those stated in the previously edited MLC.log file. The index in MLC.log file mean the cumulative radiation

**Table 3. Relative doses at the first and last segments in relation to the overshoot effect after the three corrections were applied.**

|  |  | No Correct | Block | Reverse | Index |
|---|---|---|---|---|---|
| 1MU | 1st segment | 152.5% | 95.1% | 104.8% | 100.1% |
|  | 5th segment | 48.6% | 97.3% | 91.1% | 95.9% |
| 2MU | 1st segment | 114.2% | 97.6% | 101.2% | 100.5% |
|  | 5th segment | 87.4% | 95.8% | 92.9% | 98.8% |
| 3MU | 1st segment | 114.2% | 97.9% | 98.5% | 99.0% |
|  | 5th segment | 87.4% | 95.8% | 94.2% | 98.7% |

occupation of the segments. These two principles make editing the MLC.log file difficult for all three corrections, especially with respect to the treatment fields. A program can be used to edit the MLC.log file, and the treatment fields can be tested with the corrections during the dose agreement, as performed in Zhen et al (2016).

## Conclusions

The overshoot effect on dose is more obvious for step-and-shoot IMRT, which has only a few segments. The three corrections of this study mitigated the overshoot effect on dose. To save time and effort, the MLC.log file should be edited with a program in the future.

## Supporting information

**S1 Fig. The step-and-shot MLC position were designed five segments.**
(TXT)

**S2 Fig. Relative dose measured at various MU values and dose rates.**
(XLSX)

**S3 Fig. Relative dose of three corrections at various MU values and dose rates.**
(XLSX)

**S1 Table. Relative doses at the first segment at various MU values and dose rates.**
(XLSX)

**S2 Table. Relative doses at the last segment at various MU values and dose rates.**
(XLSX)

**S3 Table. Relative dose of three corrections at the first and the last segments.**
(XLSX)

## Author Contributions

**Conceptualization:** Chun-Yen Yu, Yih-Chyang Weng, Ching-Han Hsu.

**Data curation:** Chun-Yen Yu, Shih-Wen Wan, Yih-Chyang Weng.

**Formal analysis:** Chun-Yen Yu.

**Funding acquisition:** Shih-Wen Wan.

**Investigation:** Chun-Yen Yu.

**Methodology:** Shih-Wen Wan, Ching-Han Hsu.

**Resources:** Shih-Wen Wan, Yih-Chyang Weng.

**Software:** Shih-Wen Wan, Yih-Chyang Weng.

**Supervision:** Ching-Han Hsu.

**Writing – original draft:** Chun-Yen Yu, Ching-Han Hsu.

**Writing – review & editing:** Ching-Han Hsu.

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
