## [Decision Letter · Decision Letter 0]

18 Sep 2020

PONE-D-20-04677

Three corrections for overshoot effect improved the dose for step-and-shoot intensity-modulated radiation therapy.

PLOS ONE

Dear Dr. Hsu,

Thank you for submitting your manuscript to PLOS ONE. After careful consideration, we feel that it has merit but does not fully meet PLOS ONE’s publication criteria as it currently stands. Therefore, we invite you to submit a revised version of the manuscript that addresses the points raised during the review process.

We look forward to receiving your revised manuscript.

Kind regards,

Jerry Chun-Wei Lin

Academic Editor

PLOS ONE

Journal Requirements:

Reviewers' comments:

Reviewer's Responses to Questions

**Comments to the Author**

1. Is the manuscript technically sound, and do the data support the conclusions?

Reviewer #1: Yes

2. Has the statistical analysis been performed appropriately and rigorously? 

Reviewer #1: Yes

3. Have the authors made all data underlying the findings in their manuscript fully available?

Reviewer #1: Yes

4. Is the manuscript presented in an intelligible fashion and written in standard English?

Reviewer #1: Yes

5. Review Comments to the Author

Reviewer #1: This is an interesting topic. The authors measured and listed the corrections for overshoot effect in step-and-shoot IMRT delivery.

Overshoot effect (multiple segments per field) cannot be caught from end effect test (one segment per field) in machine QA. And clinical physicists should know the limits of the machines when applying optimization parameters in TPS.

Clinically, there are two ways to avoid or reduce overshoot effect in most TPS by

1. increasing minimum segment MU and

2. limiting total number of segments per beam (increase minimum segment MU) during plan optimization.

Probably it can be mentioned in discussion.

For clinical plans which have multiple fields, very few segments, and low MUs, overshoot effect might be significant and lower GAMMA passing rate might be found.

Typo: line 135: should be “MU/min”, not “MJ/min”

6. PLOS authors have the option to publish the peer review history of their article (what does this mean?). If published, this will include your full peer review and any attached files.

Reviewer #1: No

---

## [Author Response · Author response to Decision Letter 0]

4 Nov 2020

When submitting your revision, we need you to address these additional

requirements.

1. Please ensure that your manuscript meets PLOS ONE's style

requirements, including those for file naming.

ANS: Yes, we did.

2. We suggest you thoroughly copyedit your manuscript for language

usage, spelling, and grammar. If you do not know anyone who can help you

do this, you may wish to consider employing a professional scientific

editing service.

ANS: Yes, we did. As the attached file, ***

Reviewer's Responses to Questions

Comments to the Author

1. Is the manuscript technically sound, and do the data support the

conclusions?

Reviewer #1: Yes

ANS: Thank Reviewer.

2. Has the statistical analysis been performed appropriately and

rigorously?

Reviewer #1: Yes

ANS: Thank Reviewer.

3. Have the authors made all data underlying the findings in their

manuscript fully available?

Reviewer #1: Yes

ANS: Thank Reviewer.

4. Is the manuscript presented in an intelligible fashion and written in

standard English?

Reviewer #1: Yes

ANS: Thank Reviewer.

5. Review Comments to the Author

Reviewer #1: This is an interesting topic. The authors measured and

listed the corrections for overshoot effect in step-and-shoot IMRT

delivery.

ANS: Thank Reviewer.

Overshoot effect (multiple segments per field) cannot be caught from end

effect test (one segment per field) in machine QA. And clinical

physicists should know the limits of the machines when applying

optimization parameters in TPS.

ANS: We agree.

Clinically, there are two ways to avoid or reduce overshoot effect in

most TPS by

1. increasing minimum segment MU and

2. limiting total number of segments per beam (increase minimum segment

MU) during plan optimization.

Probably it can be mentioned in discussion.

ANS: Thank Reviewer a lot. We did in new manuscript.

For clinical plans which have multiple fields, very few segments, and

low MUs, overshoot effect might be significant and lower GAMMA passing

rate might be found.

ANS: We agree.

Typo: line 135: should be "MU/min", not "MJ/min"

ANS: Thank Reviewer a lot. I changed "MJ/min" to "MU/min" at line 135 in new manuscript.

6. PLOS authors have the option to publish the peer review history of their article (what does this mean? [3]). If published, this will include your full peer review and any attached files. 

Do you want your identity to be public for this peer review? For information about this choice, including consent withdrawal, please see our Privacy Policy [4]. 

Reviewer #1: No

ANS: We respect to Reviewer’s option.

---

## [Decision Letter · Decision Letter 1]

23 Dec 2020

PONE-D-20-04677R1

Three corrections for overshoot effect improved the dose for step-and-shoot intensity-modulated radiation therapy.

PLOS ONE

Dear Dr. Hsu,

Thank you for submitting your manuscript to PLOS ONE. After careful consideration, we feel that it has merit but does not fully meet PLOS ONE’s publication criteria as it currently stands. Therefore, we invite you to submit a revised version of the manuscript that addresses the points raised during the review process.

ACADEMIC EDITOR: 

Please check the reviewer's comments to revise the manuscript

We look forward to receiving your revised manuscript.

Kind regards,

Jerry Chun-Wei Lin

Academic Editor

PLOS ONE

Reviewers' comments:

Reviewer's Responses to Questions

**Comments to the Author**

1. If the authors have adequately addressed your comments raised in a previous round of review and you feel that this manuscript is now acceptable for publication, you may indicate that here to bypass the “Comments to the Author” section, enter your conflict of interest statement in the “Confidential to Editor” section, and submit your "Accept" recommendation.

Reviewer #1: All comments have been addressed

2. Is the manuscript technically sound, and do the data support the conclusions?

Reviewer #1: Yes

3. Has the statistical analysis been performed appropriately and rigorously? 

Reviewer #1: Yes

4. Have the authors made all data underlying the findings in their manuscript fully available?

Reviewer #1: Yes

5. Is the manuscript presented in an intelligible fashion and written in standard English?

Reviewer #1: Yes

6. Review Comments to the Author

Reviewer #1: From clinical plan perspective, not every step-and shoot IMRT plan should be corrected. Can author address more on the examples/plans that impacted by overshoot effect. For example, a publication, The step‐and‐shoot IMRT overshooting phenomenon: a novel method to mitigate patient overdosage. by Heming Zhen etc., found the correction can improve prostate plan QA passing rate. And another publication, Clinical implications of the overshoot effect for treatment

plan delivery and patient-specific quality assurance for step-and-shoot IMRT. by John A. Baines etc., compared prostate and head and neck plans.

QA passing rate can be improved with these corrections. But is it necessary to apply the correction for all step-and-shoot plan? Or if other methods used during plan optimization, like increasing minimum segment MU or reducing dose rate, can the correction can be waived?

7. PLOS authors have the option to publish the peer review history of their article (what does this mean?). If published, this will include your full peer review and any attached files.

Reviewer #1: No

---

## [Author Response · Author response to Decision Letter 1]

2 Feb 2021

Reviewer #1: From clinical plan perspective, not every step-and shoot IMRT plan should be corrected. Can author address more on the examples/plans that impacted by overshoot effect. For example, a publication, The step‐and‐shoot IMRT overshooting phenomenon: a novel method to mitigate patient overdosage. by Heming Zhen etc., found the correction can improve prostate plan QA passing rate. And another publication, Clinical implications of the overshoot effect for treatment plan delivery and patient-specific quality assurance for step-and-shoot IMRT. by John A. Baines etc., compared prostate and head and neck plans. QA passing rate can be improved with these corrections. But is it necessary to apply the correction for all step-and-shoot plan? Or if other methods used during plan optimization, like increasing minimum segment MU or reducing dose rate, can the correction can be waived?

ANS: Honestly, I have to confess that the plans with overshoot effect are much fewer, even served all cases in last year in our department. I also agree that overshoot effect was minimized as increasing segment MU or reducing dose rate. I also believe that increasing segment MU or reducing dose rate is most efficient to minimize overshoot effect. The above opinions were reported in our manuscript. 

Block correction exclude overshoot and undershoot segments. Reverse-sequence correction, and index correction are compensate overshoot or undershoot segments to each other. Theoretically, overshoot effect could be reduce to zero with the three corrections, not only minimize overshoot effect. Unfortunately, before the corrections will be applied, corrections have to be approved by government. The other problem is time consuming for calculating the index of each segment and re-arranging the order of all segment. A computer program have to design for the problem.

I have to emphasize that the purpose in this report is evaluated and analyzed the availability of the three correction before the corrections are applied for clinical cases. I also have to emphasize that increasing segment MU or reducing dose rate is most efficient to minimize overshoot effect. I also believe that the result of overshoot effect as increasing segment MU or reducing dose rate is acceptable in clinical.

---

## [Decision Letter · Decision Letter 2]

5 Apr 2021

Three corrections for overshoot effect improved the dose for step-and-shoot intensity-modulated radiation therapy.

PONE-D-20-04677R2

Dear Dr. Hsu,

We’re pleased to inform you that your manuscript has been judged scientifically suitable for publication and will be formally accepted for publication once it meets all outstanding technical requirements.

Kind regards,

Jerry Chun-Wei Lin

Academic Editor

PLOS ONE

Additional Editor Comments (optional):

Authors have addressed all the queries thus the paper is acceptable for the publication now

Reviewers' comments:

Reviewer's Responses to Questions

**Comments to the Author**

1. If the authors have adequately addressed your comments raised in a previous round of review and you feel that this manuscript is now acceptable for publication, you may indicate that here to bypass the “Comments to the Author” section, enter your conflict of interest statement in the “Confidential to Editor” section, and submit your "Accept" recommendation.

Reviewer #1: All comments have been addressed

Reviewer #2: All comments have been addressed

2. Is the manuscript technically sound, and do the data support the conclusions?

Reviewer #1: Yes

Reviewer #2: Yes

3. Has the statistical analysis been performed appropriately and rigorously? 

Reviewer #1: Yes

Reviewer #2: Yes

4. Have the authors made all data underlying the findings in their manuscript fully available?

Reviewer #1: Yes

Reviewer #2: Yes

5. Is the manuscript presented in an intelligible fashion and written in standard English?

Reviewer #1: Yes

Reviewer #2: Yes

6. Review Comments to the Author

Reviewer #1: Several correction techniques have been measured and compared for the overshoot effect in this article for step-and-shoot IMRT. Authors also well explained the measurement methods to distinguish the correction methods.

Reviewer #2: Please consider improving your reference list as it is very short, some suggestions:

Liu S, et al. Property of self-similarity between baseband and modulated signals. Mobile Networks and Applications. 2019 Sep 10:1-1.

Singh K, et al. Local Statistics-based Speckle Reducing Bilateral Filter for Medical Ultrasound Images. Mobile Networks and Applications. 2020 Dec;25(6):2367-89.

Połap D, et al. Neural image reconstruction using a heuristic validation mechanism. Neural Computing and Applications. 2020 Jun 20:1-1.

7. PLOS authors have the option to publish the peer review history of their article (what does this mean?). If published, this will include your full peer review and any attached files.

Reviewer #1: No

Reviewer #2: No

---

## [Editor Report · Acceptance letter]

13 Apr 2021

PONE-D-20-04677R2 

Three corrections for overshoot effect improved the dose for step-and-shoot intensity-modulated radiation therapy 

Dear Dr. Hsu:

I'm pleased to inform you that your manuscript has been deemed suitable for publication in PLOS ONE. Congratulations! Your manuscript is now with our production department. 

Kind regards, 

on behalf of

Prof. Jerry Chun-Wei Lin 

Academic Editor

PLOS ONE